# Improved Music Perception after Music Therapy Following Cochlear Implantation in the Elderly Population

**DOI:** 10.3390/jpm12030443

**Published:** 2022-03-11

**Authors:** Astrid Magele, Bianca Wirthner, Philipp Schoerg, Marlene Ploder, Georg Mathias Sprinzl

**Affiliations:** 1University Clinic St. Poelten, Department of Otorhinolaryngology, Head & Neck Surgery, Dunant-Platz 1, 3100 St. Poelten, Austria; bianca.wirthner@stpoelten.lknoe.at (B.W.); philipp.schoerg@stpoelten.lknoe.at (P.S.); marlene.ploder@stpoelten.lknoe.at (M.P.); georg.sprinzl@stpoelten.lknoe.at (G.M.S.); 2Karl Landsteiner Institute of Implantable Hearing Devices, 3100 St. Poelten, Austria

**Keywords:** elderly, response to therapy, cochlear implantation, music therapy, rehabilitation, music perception, quality of life

## Abstract

Background: Cochlear implantation (CI) and the accompanying rehabilitation has become a routine procedure in hearing restoration. Literature is sparse on elderly CI recipients focusing on the issue of age and their inclined auditory resolution, taking their diminished cognitive function into account, which requires adaptation of rehabilitation programs to overcome habituation. Objective: This study aims to show that a few adjustments in the therapy program towards age, mental, physical and auditory condition significantly improve music perception and overall auditory benefit, hence normal communication and social interactions can be found. Methods: Subjects implanted with a CI 65 years or older were compared to age-matched normal hearing subjects. Questionnaires were administered before and after ten music therapy sessions, to evaluate the participant’s music habits, the perception of sound quality and self-awareness and hearing implant satisfaction. Results: The greatest benefit was seen in participants’ gain in self-confidence and enjoyable music perception. Not only did the amount of listening to music increase, but also the impression of sound quality changed from poor up to good/very good sound quality. Conclusions: The music therapy was well accepted and resulted in beneficial subjective as well as objective outcomes towards hearing and music impression, hence improved quality of life.

## 1. Introduction

Cochlear implantation (CI) has become a routine procedure in the hearing rehabilitation of pre- and post-lingual deaf children as well as post-lingual deaf adults. Over the years, technical advances have improved the quality of speech understanding and speech perception further. Nevertheless, CI recipients, especially the post-lingually deaf population, despite experiencing a very good speech understanding, describe listening and enjoying music as problematic. It is well accepted that it requires training and rehabilitation to overcome a certain habituation effect until music is enjoyable [1,2,3,4,5,6]. For those patients, music therapy (MT) is highly recommended, and many articles report the positive effect of MT in rehabilitation programs, resulting in significant improvements in music perception in hearing-impaired listeners [5,7,8,9,10]. Literature is still sparse on studies including elderly CI recipients focusing on the issue of age and the auditory resolution [4,7]. Huber et al. demonstrated in a case-control study widespread cognitive impairment in elderly patients with severe hearing impairment when compared to a normal hearing matched control group [11]. Cortical plasticity correlates with age, hence the listening task also influences the effectiveness of auditory rehabilitation [2]. Sarant et al. reported a significant improvement in cognition already after 18 months of appropriate auditory rehabilitation in 97% of the patients [12]. One-third of the CI patient cohort at our clinic is older than 65 years of age and the aforementioned results correspond exactly with our experience in the rehabilitation work and MT sessions with elderly participants. The elderly patient cohort usually requires more time to get used to the new hearing situation, necessitating more auditory as well as device handling training sessions. Most of the users are in a good general medical condition and have high expectations and request to keep their standard in terms of quality of life (QoL), which, on an objective/measurable level, translates into good results in speech understanding and good music perception.

Therefore, we focused our evaluation on the elderly population, to show the positive effect of MT as a complementary rehabilitation tool. Since MT supports the correlation of age and cognitive declines, it may assist the elderly in their music perception, hearing rehabilitation, regaining mental and physical stability, hence improving QoL.

The aim of this study was to demonstrate the effects of MT measured at music perception and self-awareness for elderly CI users (older than 65 years of age) and to develop a test battery which is easy to handle for this group. The outcomes were compared with each patient serving as his own control (best-aided before and after MT) and compared to a normal hearing reference group (control) to identify details about the quality and difficulty of the music test developed.

## 2. Material and Methods

This prospective study was conducted between 2017 and 2019 at the Department of Otorhinolaryngology, Karl Landsteiner Private University Clinic Hospital St. Poelten.

The study was approved by the Ethics Committee of the Federal State Lower Austria, St. Poelten (GS1-EK-4/495-2017). CI recipients older than 65 years of age, uni- or bilaterally implanted for at least 6 months with a daily regular wearing time of their speech processor of at least 8 h per day, were included. CI recipients with single-sided deafness were excluded.

### 2.1. Study Cohort

Between 2017 and 2018, a total of 231 people were implanted with a CI. Nearly half of the population (*n* = 97) were 65 years or older at time of surgery. All patients were offered, as part of the clinical rehabilitation program, to participate in 10 consecutive MT sessions. Note that 57% of the implanted cohort over the age of 65 agreed to take part in the MT sessions; the remaining 43% refused, mainly due to personal reasons and time constraints. A few declined as they were not interested in this form of therapy, especially since they were satisfied with their new hearing impressions.

The remaining CI recipients (57%) attended the MT sessions on a regular basis, out of which 11 were willing to participate in this study.

Ten patients were implanted with a Med-El (Synchrony implant and OPUS2 or Sonnet audio processors (AP) (MED-EL GesmbH, Innsbruck, Tyrol, Austria) and one with Cochlear (CI522/Kanso AP) (Cochlear Ltd., Sydney, NSW, Australia).

Each participating subject was scheduled for a 50 min MT session per week (total therapy time frame 10 weeks) and the content of each session is described in Table 1.

To evaluate information regarding the participant’s music habits, the perception of sound quality and self-awareness, two questionnaires (Munich Music Questionnaire (MUMU), Hearing Implant Sound Quality Index (HISQUI)) and one Visual Analogue Scale (VAS) as well as the music perception test (MWT) were administered. After completion of the ten MT sessions, the CI recipients were interviewed, to evaluate their emotional expressions, their subjective experiences and their own subjective impression therapeutic process in addition to the afore mentioned questionnaires.

A cohort of normal hearing subjects served as a control group (NH) (aged 65 years or older) and received the HISQUI, MUMU and MWT questionnaires once. The MUMU Questionnaire collects actual data about music listening habits of CI users [13]. For the NH-cohort, a short version of the questionnaire (23 out of 46) was adapted to exclude hearing device-specific questions.

### 2.2. Pre- and Post-Design (before and after 10 MT Sessions)

(a)MUMU Questionnaire: developed by S.J. Brockmeier, (MED-EL) is a tool to scientifically collect actual data about music listening habits of CI users with post-lingual deafness [13]. For the NH-cohort a short version of the questionnaire (23 out of 46) was adapted to exclude hearing device-specific questions.(b)HISQUI Questionnaire: was established for adults concerning subjective sound quality detection after CI surgery (MED-EL) [14]. The HISQUI measures the sound quality in everyday life situations. The total HISQUI score is obtained by adding the numerical values of all 29 evaluated questions. The score achieved overall indications of how good or poorly you find the sound quality in your personal everyday listening situations with the hearing implant. The result is interpreted by a score of the total achieved numbers: Very poor sound quality < 60, poor sound quality 60–90, moderate sound quality 90–120, good sound quality 120–150 and very good sound quality 150–203.(c)VAS: The VAS serves as evaluation of the effects of MT on self-confidence, social-participation, actual well-being, frustration, motivation, quality of life, confidence, acceptance on a scale from 0 to 10 [15].(d)MWT: The music perception test is an objective method, to test different parameters in music skills containing the following musical aspects: detection of sound sequences (descending, rising or constant), pitch discrimination, differentiation of one or two notes (unison vs. polyphony), rhythm, discrimination and recognition of different instruments. The MWT is divided into two parts: the first part (melody, pitch, rhythm, unison or polyphony) is tested by playing live on the harp by the music therapist. The harp was used because, within therapeutic sessions, it turned out that this instrument produced “enjoyable” sound and offered enough pitch range. The second part, the instrument recognition, was investigated by playing solo versions of professional artists using two loudspeakers.(e)Interviews: after completion of the 10 MT sessions, interviews were conducted by the music therapist and recorded with a Zoom H1 Handy Recorder. All interviews were transcribed and evaluated per qualitative criteria using Strauss and Corbin’s Grounded Theory model [16].

The outcomes were evaluated applying qualitative criteria using the Strauss and Corbin’s Grounded Theory model [16].

One MT session lasted for 50 min and was divided into three parts based on the therapeutic levels (emotional, functional and musical level), which were individually adapted according to the patient’s current psycho-emotional state and needs.

## 3. Data Analysis

The statistical analysis was conducted using GraphPad Prism version 5.00 for Windows, (GraphPad Prism version 8.0.0 for Windows, San Diego, CA, USA). Non-parametric Mann–Whitney t-test was used to test for significant differences between the test conditions (pre-MT, post-MT and NH) and individual analysis for uni- and bilateral users was performed. Correlation analysis was performed regarding the duration of HL (years) or experience with the CI (months) and the calculated benefit of the several questionnaire dimensions (i.e., Frustrations, Motivation, Self-confidence, Participation, etc.). Descriptive statistics were performed for age, gender, CI experience, etc., and is summarized in Table 2. Individual outcomes of the questionnaires are summarized in Table 3 and Table 4.

## 4. Results

Demographic details can be found in Table 2. The mean age of the CI group was 72.8 ± 5.7 years and 71.8 ± 4.9 years in the NH group (*p* = 0.332). The duration of using the speech processor varied from 6 to 144 months (41.1 ± 43.6/23.7 ± 19.8 months le/re), the range of duration of deafness within the group was between 3 and up to 20 years (18.25 ± 19.50 years).

The MUMU questionnaire contains 25 questions; here summarized are only the ones where the authors expected an impact of MT. Questions such as “What kind of music do you listen to”, “If you sing, indicate where”, “If you sing, please indicate what” were not statistically evaluated. With regards to the results on how long the participants listened to music, it was noticed that there was no significant difference before HL was established, when the hearing loss became established and after the patients received a CI (*p* = 0.629), neither did the outcomes significantly differ among NH subjects (*p* = 0.123). Even though, it appeared when comparing the mean values as if the patient reduced perceiving and showing interest in listening to music slightly with onset of HL, which positively increased again after hearing rehabilitation with a CI (mean: 5.36 ± 3.38, 4.56 ± 3.80 and 4.91 ± 3.18, respectively (*p* = 0.800). NH mean: 7.0 ± 2.75 (*p* = 0.313)). The role of music in the life of the participants was not significantly different in any of the stages, nor compared to the NH group (*p* = 0.085). The amount of time spent listening to music was significantly different before the onset of HL and with HL present but untreated (*p* = 0.02); no difference was found after CI treatment (*p* = 0.441) (Figure 1).

Before MT and before the onset of HL, 27% of the subjects listened to music less than 30 min and between 30 min and 1 h per day, and 9% listened 1 to 2 h per day and 37% for more than 2 h per day. No one listened to music all day. With the onset of HL, the amount of time spent listening to music changed to 73% and 18% listening for 30 min and between 30 min and 1 h, respectively. The number of listeners between 1 and 2 h remained the same (9%). No one listened to music for more than 2 h (changed from 37% to 0%) or all day. After receiving the CI, the amount of time changed positively: 55% reported less than 2 h per day, 9% between 30 min and 1 h, 27% listened to music between 1 and 2 h and 9% for more than two hours. Still, no one listened to music all day. After the 10 sessions of MT, 46% of the subjects listened to music for less than 30 min; 18% reported 30 min to 1 h; also, 18% listened to music between 1 and 2 h, 9% listened for 2 h or all day, respectively. This change towards an equal distribution of amount of time spent listening to music is comparable to the NH group, with the exception that no one listened to music all day.

When we asked ‘‘Why do you listen to music?’’, after MT, 100% of the subjects listened to music to relax, followed by “for pleasure” (88%), to improve their mood (44%), and only one subject out of nine to stay awake and/or to dance (11%) (multiple answers possible). No one listened to music for professional reasons. Before MT, the order was similar. After MT, 73% were able to identify pleasant tones (before MT, 64%), whereas all subjects could identify unpleasant sounds (before MT, 27%). Furthermore, all subjects (100%) could detect rhythm and 82% melody (before MT, 100% and 55%, respectively).

In all groups, participants were able to distinguish between high and low notes.

The subjects reported that they started again listening to music on a regular basis after receiving their CI with a mean of 1.2 months (range 0.25–3 months), one person could not remember exactly. In addition, 80% worked on music listening during their rehabilitation, 60% listened to familiar music followed by listening to unfamiliar music or played familiar music repeatedly without reading the music. No one took music lessons nor listened to and read music.

Investigating the questions of the Visual Analog Scale (Table 3), the CI group rated their “frustration of their hearing” after MT was significantly reduced (*p* = 0.0097). The category “motivation” was not significantly different when comparing pre- to post-MT (*p* = 0.341). Also, the self-confidence before MT and after MT was better afterwards, despite not being significantly different (*p* = 0.356). Their social inclusion and participation significantly improved after MT (*p* = 0.017). Well-being, quality of life, acceptance and hearing perception were rated in the positive upper third (between 7 and up to 9 out of max 10) but did not change significantly after MT (*p* > 0.05).

Evaluating the HISQUI outcomes grouped into the respective categories of sound quality, before MT, a mean of 60 ± 21.8, which corresponds to poor sound quality, was found. After the MT sessions, the mean significantly increased to 74.2 ± 27.5 (Figure 2, Table 3) for the moderate/average sound quality group (*p* = 0.021).

Additionally, the HISQUI as a function of the duration of the CI use was investigated, showing a significant difference in sound quality impression in the experienced group with more than 12 months of CI use compared to under 12 months of device use (*p* = 0.026) (Figure 3).

No significant difference was observed in any of the questionnaires and the VAS for pre-MT, post-MT and NH, except the use of music instruments, which was significantly different in the NH group when compared to pre-MT (Table 3).

The outcomes of the MWT in the NH showed that the sound sequences were correctly recognized in 95%, with a mean of 5.7 ± 0.9 correct answers out of 6, which in the CI group before MT was 4.8 ± 0.9 and improved after MT to 5.5 ± 0.7 (*p* = 0.1048).

The pitch discrimination was correct in the NH group with a mean of 4.7 ± 0.9, which was significantly different before MT (*p* = 0.0235). This outcome significantly improved in the CI group from 3.2 ± 1.4 to 4.4 ± 1.2 after MT (*p* = 0.020) (Figure 4 and Table 4).

Differentiation of one/two notes (unison vs. polyphony) was measured in the NH group with a mean of 4.9 ±1.2, which was not significantly different compared to the CI group before and after MT (4.2 ± 1.4 and 5.0 ± 1.0, respectively; *p* = 0.053) (Figure 4 and Table 4).

Interestingly, the rhythm correctly repeated was the same after MT when compared to the NH group, but was not significantly different compared to before MT.

The NH group could significantly differentiate between instruments with a mean score of 4.7 ± 1.2 when compared to after MT, with a mean of 3.3 ± 1.3 (*p* = 0.049). Duration of HL (years) as well as CI experience (months) was independent of any outcomes evaluated (correlation analysis: benefit outcomes (pre-MT–post-MT) for all questionnaire dimensions).

Finally, the interviews following the last MT session showed that the expectations of 10 out of 11 participants were fulfilled. The analytic process emerged 3 main categories: improved subjective music perception, importance of the therapeutic talks and variety of methods. Positive comments concerned: Tips for at-home exercises, pitch discrimination exercises, musical games, differentiation of sounds, noises and music pieces, playing music in an active way, discovering the joy of music, exploring new instruments, musical exercises with lyrics, the therapeutic conversation, the psychological support and the feeling of being understood. Of note, 10 out of 11 succeeded in transferring the exercises to their everyday life. To sum up, 10 out of 11 participants experienced a subjective improvement in music perception and confirmed that they gained more self-confidence in their daily life. All participants stated that they would continue professional MT sessions if they believed that it would further improve the appreciation of music and speech understanding in their daily life conditions.

## 5. Discussion

With the aging population, the trend towards a higher age at time of CI surgery is also constantly growing (our oldest CI patient was 92 years of age at implantation). Emphasis in terms of special training and rehabilitation needs in the elderly population needs to be drawn. There is a clinical need to integrate the elderly in a special rehab program and in the scientific work. Huber et al. described in their study a widespread cognitive impairment in elderly patients with hearing loss compared to a matched control group [11]. This may be challenging in the rehab work with elderly CI users, and requires special adaption of the training program.

The willingness of nearly 60% of the CI recipients to join the MT sessions shows the motivation of the patients to work professionally on their new gained hearing. Only a few rejected the offer, the reason being that they were already satisfied with their new hearing situation. There is a clear recommendation for the use of music therapy during the rehabilitation process for children and adults with CI in the context of speech and language therapy [10]. The current study aimed to show that this is also not only applicable, but important in the elderly group of patients, aged 65 years or older.

There are a lot of test batteries, which are used to include music skills, musical preferences and quality-of-life facts. The MUMU was especially created for the adult population with post-lingual CI users to assess the aspect of music appreciation [17]. We integrated MUMU in our test concept as one of the subject music appreciation measurements, next to the HISQUI and the VAS.

The MUMU questionnaire revealed that the time of listening to music did increase after MT, but did not exceed the amount of time the participants listened to music before the HL occurred. The interest to listen to music appeared significantly after MT. One explanation could be the improvement of the sound quality, which we evaluated with the MUMU. The reverberant sound when listening to music could be significantly reduced after MT. It was easier for the study participants to identify unpleasant and pleasant sounds and recognize rhythm and melodies. So, we can conclude that MT can bring back the motivation to listen to music. This influences their hearing rehabilitation positively and supports their endurance in their training.

Inspired by Hutter, who used the Multidimensional Self-Concept scale (MSCS) to provide assessment data relating to global self-esteem and the six content-dependent self-esteem subscale [15], we developed our own Visual Analog Scale (VAS) to obtain data about self-awareness before and after MT. We paid attention to make it easy and uncomplicated to handle for the investigated elderly group and to limit the time frame for completing all questionnaires.

An improvement after MT was also seen in the VAS. The frustration of their hearing was significantly reduced, and their social inclusion and participation significantly improved. Consequently, their former social contacts were refreshed and more social activities were taken up again. To widen your social field means more training in natural conditions by speaking and listening to your environment. Every kind of training is important for improvement in speech understanding; the more the CI users practice, the more naturally the new electrical hearing becomes.

The HISQUI results showed that all of the CI users rated their hearing before MT with poor sound quality, which significantly improved after MT sessions. Three study participants even reached the same level of very good sound quality, just like the NH group. Interestingly, all three were bilaterally implanted, which underlines, once more, that also stereo conditions are a success factor of hearing rehabilitation with CI. Especially, elderly people often refuse a second implant (bilateral), due to possible side effects of a second surgery. The outcomes are once more in agreement with the literature, that two CIs improve directional hearing and speech understanding, but furthermore, also sound quality and sound discrimination regarding music perception is better when bilaterally treated [18]. Although Peterson claims that the effect of music training on speech perception, especially in the long-term results, is unknown [19], we found out that all long-time experienced users (>12 months) exhibited a significantly better sound quality outcome compared to the under-12-month (<12 months) users. So, we assume that a longer time of MT can also bring more improvement in sound quality and should be integrated in the routine rehab program. This statement is consistent with the recommendation of Shukor in the literature review, where they suggest a long training duration (12 months or longer) to optimize the effectiveness of rehabilitation programs for hearing-impaired individuals [20]. Shukor et al. confirmed in a meta-analysis the importance of long duration of training [20].

The interview analysis revealed overall positive feedback from all candidates. The willingness to transfer the training into their everyday life shows that all of them had experienced a personal benefit, which they wish to expand in the future. The biggest benefit was described in the improvement of music perception, which brings back more quality of life. Also, the increase of their self-confidence leads to more social contacts, which helps them to enhance in real-life conditions. The participants recognized the positive side effect of MT in terms of better speech understanding and all were willing to continue the therapy in the future. MT improved not only the peripheral hearing, but also the central auditory processing and cognition, which led to better comprehension of speech, especially in the elderly.

Altogether, we can conclude that we can highly recommend MT for CI users older than 65 years. MT brings not only an improvement in quality of life, but it can also support the speech understanding rehabilitation and helps to bring them, due to higher self-esteem and more self-confidence, back to social life. MT should not be missing in any rehab program for CI users, whether for children, adolescents, nor the elderly population.

## Figures and Tables

**Figure 1 jpm-12-00443-f001:**
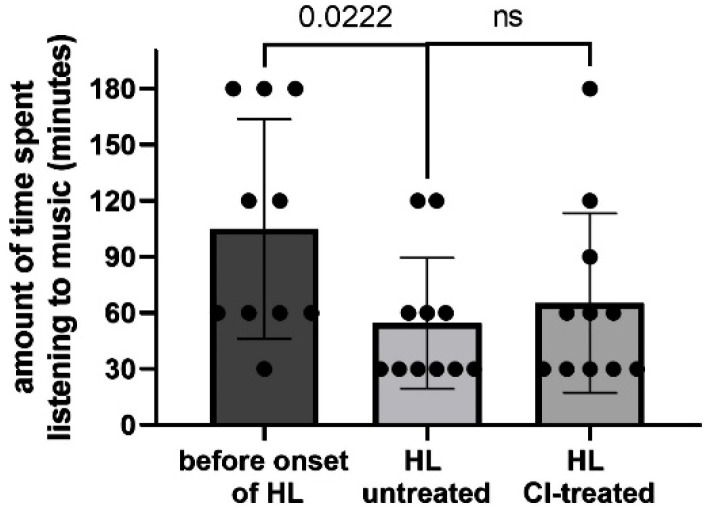
Distribution of amount of time of patients listening to music before onset of hearing loss (HL), when HL occurred with treatment and after CI rehabilitation and Music Therapy.

**Figure 2 jpm-12-00443-f002:**
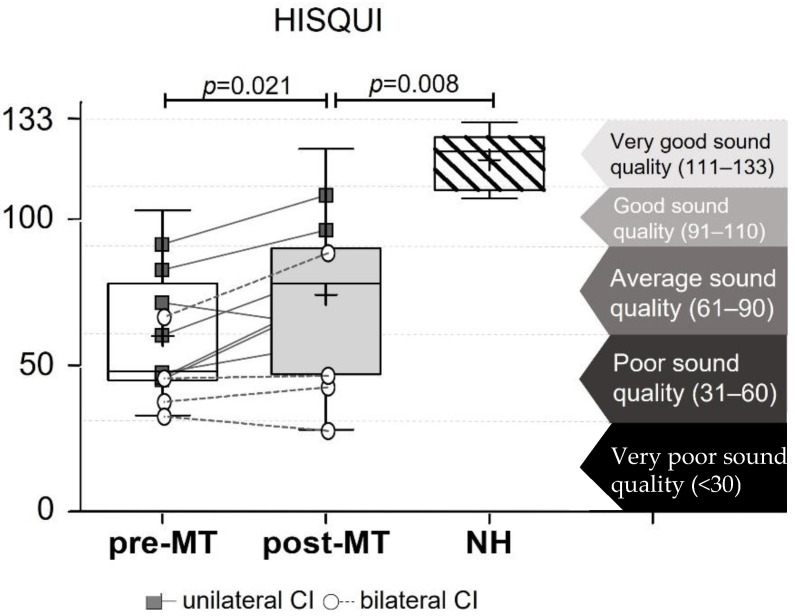
Hearing Implant Sound Quality Index separated into the four sound quality groups, comparing the outcomes before and after Music Therapy.

**Figure 3 jpm-12-00443-f003:**
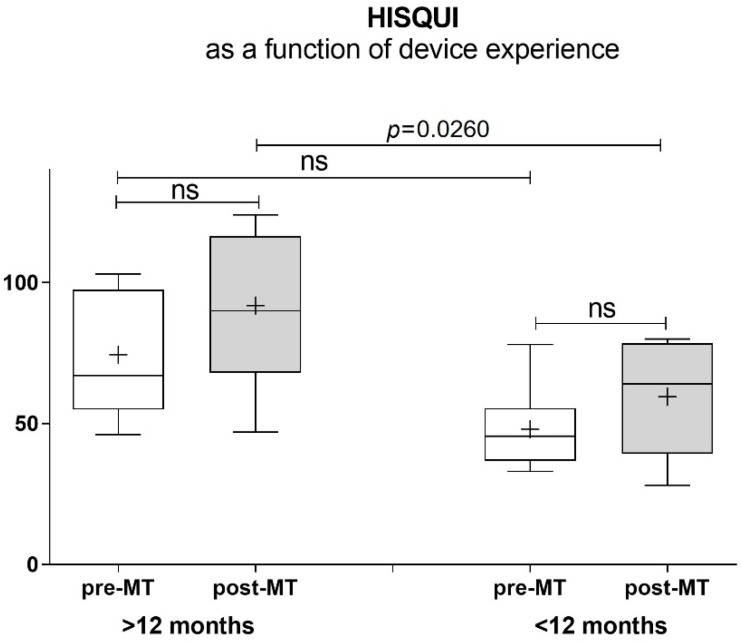
Hearing Implant Sound Quality Index as a function of device experience.

**Figure 4 jpm-12-00443-f004:**
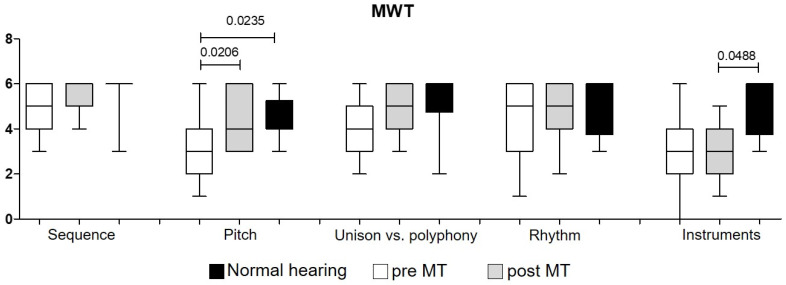
The music perception test (MWT) evaluating different parameters in music skills containing the following musical aspects: detection of sound sequences, pitch discrimination, differentiation of one or two notes (unison vs. polyphony), rhythm, discrimination, and recognition of different instruments.

**Table 1 jpm-12-00443-t001:** Description of the Therapy sessions.

Session	Functional Level	Therapeutic Objective
8,9,10	Speech comprehension with music in the background	Ability to focus on a communication partner in a noisy environment
1,2	Rhythmical exercises as supportive tool to understand words with different syllables	Syllable supported rhythms to improve speech perception/comprehension
5,6	Familiar songs and well-known melodies with changed lyrics	Improving concentration ability
1,3	Discrimination of Sounds and Noises	Raise awareness in hearing experiences in everyday life
1	Writing down hearing experiences in a “Hearing-Diary”	Discovery and sensible perception of every-day-life listening environment
2,3	Finding and formulating hearing strategies	Implementation of individual hearing strategies in everyday life
4	Melody recognition of nursery rhymes/ familiar songs with/wo lyrics	Training in lyrics comprehension
8,9	Nursery rhymes with close test	Speech understanding using easy songs
7	Listening and talking about classical compositions in different versions	Learning skills to explore differences in instruments, tempi, dynamics etc. in cover versions
4	Recognition of unison vs. polyphony	Training in discrimination and hearing awareness
4,5	Musical exercises with digital media	Additional digital musical training to give more exercise opportunities
2,3	Audio exercises: discrimination of voices (male/female), instruments etc.	Learning to discriminate voices and timbres
1-10	Differentiation of tone length and sequence	Exploring different sound and melody offers (raising, descending, constant)
6,7	Singing or whistling songs	Learning to differentiate sounds

**Table 2 jpm-12-00443-t002:** Patient demographics.

Subject ID	Age (Years)	Gender	Uni/ Bilateral	PTA4 CI	PTA4 Contralat.	AP Type	CI Experience (Months)	HA Contra-Lateral	Time Deafness (Years)
MTCI 01	82	F	uni/right	105	98.75	Opus 2	42	le	14
MTCI 02	69	M	uni/left	115	87.5	Sonnet	15	ri	1.5
MTCI 03	76	M	bilateral	120	120	Opus 2 (le) Sonnet (ri)	144 (le); 21 (ri)	-	14
MTCI 04	76	M	uni/right	91.25	93.75	Sonnet	8	le	70
MTCI 05	71	F	uni/right	86.25	33.75	Sonnet	24	le	3
MTCI 06	66	M	uni/left	83.75	36.25	Sonnet	6	ri	5
MTCI 07	78	M	uni/right	76.25	71.25	Sonnet	7	le	15
MTCI 08	71	F	uni/right	85	58.75	Kanso	6	le	20
MTCI 09	79	F	bilateral	68.75 (CI re)	83.75 (li)	Opus 2 (le) Sonnet (ri)	66 (le); 6 (ri)	-	15–20
MTCI 10	68	F	bilateral	113.75 (CI re)	110 (li)	Sonnet (le). Opus 2 (ri)	12 (le); 46 (ri)	-	> 20
MTCI 11	65	M	bilateral	98.75 (CI re)	71.25 (li)	Sonnet (le). Opus 2 (ri)	7 (le); 65 (ri)	-	-
MEAN/SD CI	72.8 ± 5.7	5F/6M	4 Bilat.				41.1 ± 43.6/23.7 ± 19.8	18.3 ± 19.5	
MTNH 01	65	F	-			-	-	-	-
MTNH 02	66	M	-			-	-	-	-
MTNH 03	69	F	-			-	-	-	-
MTNH 04	78	F	-			-	-	-	-
MTNH 05	75	F	-			-	-	-	-
MTNH 06	79	M	-			-	-	-	-
MTNH 07	76	F	-			-	-	-	-
MTNH 08	72	F	-			-	-	-	-
MTNH 09	71	F	-			-	-	-	-
MTNH 10	67	F	-			-	-	-	-
MEAN/SD NH	71.8 ± 4.9	7F/2M	-			-	-	-	-

AP Audio Processor; HA Hearing aid.

**Table 3 jpm-12-00443-t003:** Outcomes of the questionnaires HISQUI and VAS.

#Subjects	HISQUI		VAS							
			Frustration	Motivation	Self-Confidence	Participation	Wellbeing	QoL	Acceptance	Auditory Perception
	Pre/Post	NH	Pre/Post	Pre/Post	Pre/Post	Pre/Post	Pre/Post	Pre/Post	Pre/Post	Pre/Post
01	92/109	123	5/0	10/10	5/10	7/10	6/10	7/10	10/10	8/10
02	64/90	116	5/3	8/10	8/10	2/6	10/5	10/10	8/10	8/10
03	67/89	128	5/5	10/7	7/6	2/7	6/5	6/4	7/6	7/5
04	45/78	133	5/5	7/6	10/6	1/3	5/6	5/8	8/5	4/5
05	103/124	116	6/0	10/9	8/10	9/10	8/9	8/10	8/8	7/10
06	46/80	124	8/3	10/10	4/8	4/8	4/7	4/9	4/9	3/8
07	78/68	110	6/5	8/8	5/3	4/3	8/7	8/7	7/6	6/6
08	48/60	107	8/2	10/8	7/7	6/7	5/8	6/8	9/10	4/6
09	38/43	124	5/6	10/10	10/10	2/5	7/7	9/8	7/5	6/5
10	33/28	133	10/3	10/10	6/8	6/6	9/5	10/9	10/10	10/9
11	46/47	-	10/7	6/5	8/5	2/2	10/10	5/7	3/4	2/4
MEAN	60.0/74.2	121.4	5.7/3.6	9.0/8.45	7.1/7.6	4.1/6.1	7.1/7.2	7.1/8.2	7.4/7.6	5.9/7.1
SD	21.8/27.5	8.5	2.42/2.2	1.4/1.7	1.9/2.3	2.5/2.6	2.0/1.8	2.0/1.7	2.1/2.3	2.3/2.3
Min	33/28	107	0/0	6/5	4/3	1/2	4/5	4/4	3/4	2/4
Max	103/124	133	10/7	10/10	10/10	9/10	10/10	10	10/10	10/10

**Table 4 jpm-12-00443-t004:** Outcomes of the questionnaires MUMU and MWT.

#ID	MUMU								MWT									
	Music Perception		Pitch Differentiation Ability		Instrument		Current Listening to Music (min.)		Sequence		Pitch		Unison/ Polyphony		Rhythm		Instruments	
	Pre/ Post	NH	Pre/ Post	NH	Pre/ Post	NH	Pre/ Post	NH	Pre/ Post	NH	Pre/ Post	NH	Pre/ Post	NH	Pre/ Post	NH	Pre/ Post	NH
01	8/10	8	yes/yes	yes	no	yes	60/120	60	4/4	6	4/6	5	4/5	6	2/5	6	3/3	6
02	5/10	10	yes/yes	yes	yes	yes	30/60	120	6/6	6	3/4	6	5/5	6	3/5	6	4/4	6
03	5/6	9	yes/yes	yes	yes	no	60/60	60	5/5	3	4/4	3	6/6	5	6/5	4	3/4	3
04	5/5	10	yes/yes	yes	no	no	30/30	30	3/5	6	1/3	5	2/3	4	4/5	6	0/2	4
05	6/8	7	yes/yes	yes	no	no	120/90	60	6/6	6	6/6	5	6/6	5	6/6	3	6/5	3
06	10/8	5	yes/yes	yes	no	no	30/60	120	5/6	6	2/6	4	5/6	5	6/6	5	3/5	5
07	7/4	10	yes/yes	yes	no	no	60/30	180	5/6	6	3/3	6	4/4	6	1/4	4	2/2	6
08	5/7	10	yes/yes	yes	no	no	30/30	180	6/5	6	4/4	5	2/5	5	6/6	4	4/3	6
09	1/1	10	N/A/yes	yes	no	no	30/30	60	5/6	6	2/3	4	3/6	2	5/3	3	2/3	4
10	8/8	10	yes/yes	yes	no	no	120/180	180	4/6	6	2/4	4	4/4	5	5/2	6	1/1	4
11	21	-	yes/yes	-	no	-	30/30	-	4/6	-	4/5	-	5/5	-	6/6		4/4	-
M	5.6/6.2	8.9					54.5/65.5	105	4.8/5.5	5.7	3.2/4.4	4.7	4.2/5.0	4.9	4.5/4.7	4.7	2.9/3.3	4.7
SD	2.5/3.0	1.6					33.4/45.8	55.7	0.9/0.7	0.9	1.4/1.2	0.9	1.4/1.0	1.2	1.8/1.3	1.2	1.6/1.3	1.2
Min	1/1	5					30/30	30	3/4	3	1/3	3	2/3	2	2/2	3	0/1	3
Max	10/10	10					120/180	180	6/6	6	6/6	6	6/6	6	6/6	6	6/6	6

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
