# Peer review of "Improved Music Perception after Music Therapy following Cochlear Implantation in the Elderly Population"

_jpm, 2022, doi:10.3390/jpm12030443_

Round 1

Reviewer 1 Report

The authors assessed the music perception in a cohort of elderly after CI that underwent music therapy. A normal hearing cohort served as control group.

The title of the study is somewhat misleading as not music therapy is evaluated, but rather music perception after music therapy. If you intend to evaluate the benefit of music therapy, the control group should consist of a CI cohort that did not undergo music therapy.

Patients with unilateral or bilateral CI were included. For the patients with unilateral CI, how was the hearing on the contralateral side with and without the hearing aid?

Did you find a correlation between duration of CI training and music perception evaluated by the questionnaires?

Did the duration of deafness before CI affect the outcome of your study?

Did music perception correlate to hearing outcome?

Minor comments

Line 36: «patients» instead of «patient’s»

Consider a graphical representation of the duration of listening to music (lines 157-171).

Author Response

The authors assessed the music perception in a cohort of elderly after CI that underwent music therapy. A normal hearing cohort served as control group.

 The title of the study is somewhat misleading as not music therapy is evaluated, but rather music perception after music therapy.

Title was changes accordingly – please see Line 2/3

If you intend to evaluate the benefit of music therapy, the control group should consist of a CI cohort that did not undergo music therapy.

The initial study protocol did foresee to also evaluate a CI cohort that did not undergo music therapy. Unfortunately, the Ethics Committee of our Institution did not approve this control group as to their opinion such an evaluation would be too burdensome and time-consuming without any expected additional benefit.

Patients with unilateral or bilateral CI were included. For the patients with unilateral CI, how was the hearing on the contralateral side with and without the hearing aid?

 The Demographics table was edited accordingly – please see new table 1 and an additional audiological outcome table with the aided

Did you find a correlation between duration of CI training and music perception evaluated by the questionnaires?

Duration of training was the same for all participants. The subjects were evaluated before and after their 10 music therapy sessions (one session per week) – hence, the duration was not a useful correlation measure towards music perception. Therefore, no correlation calculations were performed for this outcome. The material and methods section was clarified regarding the timeframe of MT – please see lines 71 and 72.

What we did evaluate was a possible correlation between the prior CI experience as well as onset of hearing loss (also a Reviewer question) versus the benefit outcomes of several questionnaire measures – please see respective points below.

Correlation between the prior CI experience and the benefit in music perception, benefit HISQUI, benefit in scores of Frustrations, Motivation, Selfconfidence, Participation, Wellbeing, Quality of Life, Acceptance and Auditory perception   – the outcomes resulted in no correlations with an r squared of 0,008548; 0,01082; 0,1517; 0,2628; 0,02199; 0,1581; 0,03294; 0,2988; 0,08100 and 0,4028 respectively.

We added a sentence regarding the absence of a correlation in the results section please see lines 240-243

Did the duration of deafness before CI affect the outcome of your study?

We expected a somewhat correlation of long-lasting untreated hearing loss but outcomes did not support our hypothesis: We performed correlation analysis between duration of deafness until treatment (CI) versus the the benefit in music perception, benefit HISQUI, benefit in scores of Frustration, Motivation, Selfconfidence, Participation, Wellbeing, Quality of Life, Acceptance and Auditory perception   – the outcomes resulted in no correlations with an r squared of 0,04605; 0,02456; 0,07034; 0,06853; 0,4089; 0,02677; 0,01777; 0,02392; 0,3795 and 0,05428 respectively.

Did music perception correlate to hearing outcome?

We performed correlation analysis between hearing outcome and the benefit in music perception, benefit HISQUI, benefit in scores of Frustration, Motivation, Selfconfidence, Participation, Wellbeing, Quality of Life, Acceptance and Auditory perception   – the outcomes resulted in no correlations with an r squared of 0,08108; 0,0009879; 0,08099; 0,1646; 0,1499; 0,09280; 0,1479; 0,005075; 0,03448 and 0,001827respectively.

Minor comments

Line 36: «patients» instead of «patient’s»

Change was made accordingly please see line 37

Consider a graphical representation of the duration of listening to music (lines 157-171).

New Figure 1 was added regarding duration of listening to music

Reviewer 2 Report

It is a very interesting paper and the study design and statistical analysis are correct.

Some suggestion in order to improve the quality of the presentation and the comprehension for readers.

The 3 first periods of the results chapter should be transposed in the material and methods chapter.

The methods followed in the music therapy should be better described.

Author Response

It is a very interesting paper and the study design and statistical analysis are correct.

Some suggestion in order to improve the quality of the presentation and the comprehension for readers.

The 3 first periods of the results chapter should be transposed in the material and methods chapter.

Change was made accordingly: study cohort was added with a subheading to Material&Methods section, please see lines 72 - 83

The methods followed in the music therapy should be better described.

A summarizing table with the therapy workload per session was added (a description of the method would have been too long) please see new Table 1

The authors would have had the manuscript checked by an English native – but due to the time line given for the revision (only one week) this was unfortunately not possible – in case of acceptance of the paper the authors would like to suggest to get the final version checked by an English native.

Round 2

Reviewer 1 Report

The authors responded to all my comments. I do not have any further comments.

Reviewer 2 Report

The authors have corrected the paper according to my observations and can be published in this form.